# The C1473G Mutation in the Mouse *Tph2* Gene: From Molecular Mechanism to Biological Consequences [note 1]

**DOI:** 10.3390/biom15040461

**Published:** 2025-03-21

**Authors:** Nikita V. Khotskin, Polina D. Komleva, Alla B. Arefieva, Vitalii S. Moskaliuk, Anna Khotskina, Ghofran Alhalabi, Arseniy E. Izyurov, Nadezhda A. Sinyakova, Daniil Sherbakov, Elizabeth A. Kulikova, Daria V. Bazovkina, Alexander V. Kulikov

**Affiliations:** The Federal Research Center Institute Cytology and Genetics, Russian Academy of Sciences, Avenue Lavrentyev, 10, Novosibirsk 630090, Russia; khotskin@bionet.nsc.ru (N.V.K.); komleva@bionet.nsc.ru (P.D.K.); a.arefeva@alumni.nsu.ru (A.B.A.); vsmoskaliuk@bionet.nsc.ru (V.S.M.); dotcenko@bionet.nsc.ru (A.K.); ghofranalhalabi123@gmail.com (G.A.); izyurovae@bionet.nsc.ru (A.E.I.); sinyakova1985@gmail.com (N.A.S.); alebastr007@yandex.ru (D.S.); kulikova.elisa@gmail.com (E.A.K.); drinterf@bionet.nsc.ru (D.V.B.)

**Keywords:** serotonin, tryptophan hydroxylase 2, C1473G polymorphism, thermal stability, hind legs dystonia, behavior, lipopolysaccharide, immunity, brain, mice

## Abstract

Tryptophan hydroxylase 2 (TPH2) hydroxylates L-tryptophan to L-5-hydroxy tryptophan—the key step of 5-HT synthesis in the mammalian brain. Some mutations in the human *hTPH2* gene are associated with psychopathologies and resistance to antidepressant therapy. The C1473G polymorphism in the mouse *Tph2* gene decreases the TPH2 activity in the mouse brain. In the present paper, B6-1473C and B6-1473G congenic mice that were different only in the C > G substitution were used. The molecular mechanism of decrease in the mutant enzyme activity and some physiological and behavioral traits affected by this mutation were revealed for the first time. Analysis of thermal denaturation curves in vitro revealed that the C > G substitution reduces the free energy of denaturation, stability and lifetime of mutant TPH2. Later, we evaluated the effect of the 1473G allele on the hierarchical state, competition for a sexual partner in adult mice, mouse embryos, hind legs dystonia and the response to LPS treatment in young mice. No effect of this mutation on the hierarchical state and competition for a female was observed in adult males. The C > G substitution does not affect survival, body mass or the TPH activity in the brain of 19-day-old mouse embryos. At the same time, we found that the 1473G allele causes hind legs dystonia in juvenile (3 weeks old) mice, which can affect their escape capability in threatening situations. Moreover, a significant increase in the vulnerability to LPS in juvenile B6-1473G males was shown: a single ip LPS administration killed about 40% of young mutant mice, but not wild-type ones. The body mass of mutant males was lower compared to wild-type ones, which also can indirectly decrease their concurrent and reproductive success.

## 1. Introduction

Tryptophan hydroxylase 2 (TPH2) hydroxylates L-tryptophan to L-5-hydroxy tryptophan—the key stage of 5-HT synthesis in the mammalian brain [1,2,3,4]. *Tph2* gene knockout [5,6,7] as well as a TPH2 irreversible inhibitor, p-chlorophenylalanine [8,9], markedly decrease the 5-HT level in the mouse brain. Some mutations in the human *hTPH2* gene are associated with psychopathologies and resistance to antidepressant therapy [10,11,12,13].

Twenty years ago, the research group headed by M. Caron first discovered the C1473G polymorphism in the mouse *Tph2* gene that results in the P447R substitution in the mouse TPH2 molecule [14]. The P447R substitution lowers the TPH2 activity in brains of some mouse strains [15,16].

Although the C1473G polymorphism was intensively studied for over 20 years, there were at least two key still unsolved questions associated with this mutation. The first question concerned the molecular mechanism underlying the decrease in TPH2 activity induced by the P447R substitution. The second question concerned the physiological and behavioral mechanisms affected by the 1473G allele. Two B6-1473C and B6-1473G congenic mouse lines were created by transferring the 1473G allele encoding the low TPH2 activity from CC57Br [17], Balb/c [18,19] and AJc1 [19] mice into the C57BL/6 genome. Surprisingly, despite the marked difference in the brain TPH2 activity, these authors failed to observe any visible and reproducible difference in behavioral traits between these congenic lines in the standard behavioral tests [17,19]. However, the 1473G allele was not recognized in wild mouse populations and it was hypothesized that there was a natural selection process against this allele [20].

In this study, we tried to solve these two key questions about the P447R substitution. First, we compared the *Tph2* gene and the TPH2 protein expression in the midbrains of B6-1473C and B6-1473G mice and the kinetic and thermodynamic characteristics of wild-type and mutant enzymes in vitro. Second, we compared adult B6-1473C and B6-1473G males for their competition ability and attractiveness for females, as well as embryos and juvenile males of these genotypes for hind legs dystonia and reaction to LPS treatment.

The preliminary results of this study were presented at 14th International Multiconference Bioinformatics of Genome regulation and Structure/Systems Biology BGRS/SB [21].

## 2. Materials and Methods

### 2.1. Animals and Experimental Design

In order to minimize uncontrolled genetic and environmental variations, all experiments were carried out on B6-1473C and B6-1473G juvenile (3 weeks old) and adult (12 weeks old) males, as well as B6-1473C and B6-1473G adult females, all being homozygous for 1473C and 1473G alleles. These mice were selected from intercrosses between B6-1473CG females and B6-1473CG males (Figure 1). These experimental mice had the same genetic background and were born and brought up by B6-1473CG mothers.

Twenty days after birth, juvenile mice of the same genotype and sex were put in an Optimice cage (Animal Care Systems, Inc., Centennial, CO, USA), four animals per cage, and kept at a temperature of 22 ± 2 °C, humidity 45–50%, and artificial 14:10 (light/dark) photoperiod with daybreak and sunset at 01:00 and 15:00, respectively, until the experiments. The mice received sterile food and water *ad libitum*. All tests and sampling were carried out in the dark phase between 15:00 and 17:00.

Experiment 1: in vitro comparison of the kinetic and thermodynamic characteristics of TPH2 extracted from the midbrains of B6-1473C (n = 10) and B6-1473G (n = 10) adult males.

Experiment 2: comparison of B6-1473C and B6-1473G adult males in mixed pairs (9 pairs) for their ability to dominate in the tube dominance test.

Experiment 3: comparison of olfactory preference of B6-1473C and B6-1473G adult male urine by receptive B6-1473C (n = 19) and B6-1473G (n = 25) females in the olfactory test.

Experiment 4: comparison of B6-1473C and B6-1473G adult males in mixed pairs (22 pairs) for their ability to mate females (19 B6-1473C and 25 B6-1473G females) and pass the 1473C and 1473G alleles to their descendants.

Experiment 5: comparison of 19-day-old B6-1473C, B6-1473G and B6-1473CG embryos from B6-1473CG mothers (n = 5) for their body masses, 5-HT and 5-HIAA levels, TPH activity, and *Tph2* and *Tph1* gene expression in their brains.

Experiment 6: comparison of dystonia expression in adult (12 weeks old, B6-1473C, n = 21, and B6-1473G, n = 24) and juvenile (3 weeks old, B6-1473C, n = 16, and B6-1473G, n = 27) B6-1473C and B6-1473G males.

Experiment 7: comparison of the alterations in behavior and the brain 5-HT system in juvenile (3 weeks old) B6-1473C (n = 24) and B6-1473G (n = 29) males 5 days after single ip administration of saline and 2 mg/kg of bacterial lipopolysaccharide (LPS). Four experimental groups were formed: (1) B6-1473C saline (n = 8), (2) B6-1473C LPS (n = 8), (3) B6-1473G saline (n = 8), and (4) B6-1473G LPS (n = 12). Mice of all groups were isolated in special Phenomaster cages (TSE Systems GmbH, Berlin, Germany). After one day of adaptation, the mice in the 1st and 3rd groups were administered with saline, and mice in the 2nd and 4th groups with 2 mg/kg of LPS, and their locomotor activity and sleep were recorded for 5 successive days. On the 6th day after the treatment, all alive animals were tested in the open field test, euthanized via carbon dioxide asphyxiation followed by decapitation, their spleen and thymus were weighed, and their midbrain was rapidly dissected, frozen in liquid nitrogen and stored at −80 °C (Figure 2). In this experiment, LPS was dissolved in saline (2 mg/10 mL) and ip administered in volume 10 uL/g to the body mass.

Comment: The midbrain was chosen since it contains the bodies of all 5-HT neurons, where TPH2 expresses itself and the main amount of serotonin in the brain is synthesized.

### 2.2. Behavioral Tests

#### 2.2.1. Tube Dominance Test

This test evaluates social hierarchy in mice [22]. It was performed in a transparent plastic tube (30 cm × 3 cm). The tube diameter allows only one adult male to pass and prevents it from turning around or crawling over the other mouse.

Nine pairs consisting of one B6-1473C male and one B6-1473G male of the same age and body mass were put into nine Optimice cages. After 10 days of the habituation period, social hierarchy between the mice was established. During training, all mice went through the tube five times from each side and they knew the tube was safe. During the testing stage, mice were tested in pairs, with three trials per pair. Two mice entered the tube from opposite ends and met in the middle. The mouse that forced the opponent to back away out of the tube was designated as the dominant, the one that retreated as the subordinate. When one of the mice left the tube, the trial stopped and the time was measured. After each trial, the mice were left to rest for 2 min. The hierarchical rank of a mouse in a pair was assigned according to its dominance index—the number of wins in three trials (from 0 to 3).

#### 2.2.2. Olfactory Preference Test

This test was conducted to compare B6-1473C and B6-1473G male urine attractiveness for females. The beddings in B6-1473C and B6-1473G male home cages were not changed for 7 days. Then, these beddings soaked in B6-1473C and B6-1473G male urine were separately collected in polyethylene bags, stored at −20 °C and were used as the source of B6-1473C and B6-1473G male urine odors.

The olfactory preference test was conducted in a plastic box (90 cm × 50 cm) divided into three equal compartments (30 cm × 50 cm) separated from each other by transparent walls with small rectangular openings (Figure 3). A female was placed for 10 min in the box to get familiar with all compartments. Then, the female was locked in the central compartment, which was separated from the other two by transparent partitions, and two Petri dishes filled with bedding samples previously taken from males B6-1473C and B6-1473G were placed into two side compartments. Next, the partitions isolating the central compartment were removed simultaneously, and the female was given access to all compartments for 10 min (Figure 3).

The female movement was tracked by Microsoft Kinect 3D sensors and the EthoStudio software (Institute of Cytology and Genetics, Novosibirsk, Russia, revised in 2018) calculated the number of times and the accumulated time (s) of sniffing the bedding from the B6-1473C and B6-1473G males. After each test, the arena was carefully cleaned, and all used bedding was put in an airtight container.

#### 2.2.3. Comparison of B6-1473C and B6-1473G Adult Males in Mixed Pairs for Their Ability to Mate Females and Pass 1473C and 1473G Alleles to Their Offspring

One adult B6-1473C male and one adult B6-1473G male of the same age and body mass were placed in a population cage (35.6 cm × 48.5 cm × 21.8 cm). The large size of the cage was necessary to allow the animals to avoid each other. There was also a plastic fence in the middle of the cage with an opening in the middle. Twenty-two such mixed pairs were formed.

During a week, the males got used to the cage and established relationships in the pair. Then, one sexually mature female of the B6-1473C or B6-1473G genotype was placed for a week into each cage. A week later, the females were removed and placed back into single cages to give birth. A week after the first group of females’ weaning, the second group of females was placed into the cages with the males and the experiment was repeated, but a female with a different genotype was added to the pairs of males whenever possible. In total, the same 25 B6-1473C and 18 B6-1473G females participated in the experiment. The genotypes of their offspring were assays and the number of homo- and heterozygous litters for the B6-1473C and B6-1473G females was counted. Homo- and heterozygous litters were unambiguous evidence of the mating of a given female with a male of the same or opposite genotype.

Comment: In the experiment, we used the same females as in the experiment for odor preference.

#### 2.2.4. Tail Suspension Test

The test was conducted on two consecutive days. On each day, mice were fixed by their tails with an adhesive tape and hooked to a horizontal bar placed 30 cm above the table surface for 6 min. Dystonia was evaluated by amount and accumulated time (s) of hind legs clasping registered for 6 min after tail suspension of mice (Figure 4) [23,24].

#### 2.2.5. Home Cage Activity

Daily dynamics of locomotor activity (m) and sleep duration (hours) were recorded and calculated using a Phenomaster (TSE, Germany) according to the manufacturer’s instructions. The software determined the state of sleep as a lack of mobility for 40 s or more [25,26,27].

#### 2.2.6. Open Field Test

The mouse was placed at the wall of a brightly illuminated (300 lx) white plastic arena 55 cm in diameter bordered with a 30 cm wall and its movements were automatically tracked for 5 min [24]. The EthoStudio software automatically calculates 2 behavioral traits: (1) the time (%) spent in the central part (27.5 cm in diameter), calculated as the ratio of the animal-associated pixels in the center to the total number of animal associated pixels, and (2) the travelled distance during the test (m) [24]. The arena was cleaned with wet and dry napkins after each test.

### 2.3. Embryos

Five B6-1473CG females were mated with five B6-1473CG males. The conception beginning was diagnosed with a vaginal plug. On the 19th day of pregnancy, these pregnant females were euthanized via carbon dioxide asphyxiation and their 19-day-old embryos were weighed and genotyped, and their whole brains were frozen in liquid nitrogen and stored at −80 °C.

### 2.4. Genotyping for C1473G Polymorphism

DNA samples were isolated from the tail tips of mice using protease K treatment for 1.5 h, followed by a phenol–chloroform extraction, and diluted to the concentration of 10 ng/uL. The DNA samples were genotyped with two positive control primers and either 1473C or 1473G allele-specific primers [14,15,16,17] (Table 1). The PCR products were resolved by electrophoresis in 2% agarose gel stained with ethidium bromide.

### 2.5. Biochemical Studies

#### 2.5.1. Tissue Preparation

For the total RNA, serotonin (5-HT), 5-hydroxyindoleacetic acid (5-HIAA) and TPH2 extraction, the midbrains of adult mice and the whole brains of embryos were prepared and homogenized as has been described earlier [24]. An aliquot of 50 µL of the homogenate was used for the 5-HT and 5-HIAA assay (see Section 2.5.2) and 100 µL for the total RNA extraction (see Section 2.5.7); the rest was centrifuged and the clear supernatant was used for the TPH2 protein and activity assays (see Section 2.5.3, Section 2.5.4, Section 2.5.5 and Section 2.5.6). In order to assay the TPH2 K_M_ and V_max_ values as well as thermal stability, several midbrains of mice of the same genotypes (B6-1473C or B6-1473G) were pooled.

#### 2.5.2. Assay of 5-HT and 5-HIAA Levels

The mix of 50 μL of homogenate with 150 µL of 0.6 M HClO_4_ (Section 2.5.1) was spun for 15 min at 12,700 rpm (+4 °C). The pellet was dissolved in 1 mL of 0.1 M NaOH and used for protein determination by the Bradford method (Bio Rad, Hercules, CA, USA). The clear supernatant was twofold diluted with ultra-pure water and 5-HT and 5-HIAA levels were assayed in the diluted supernatant on a Luna C18 (2) column (5 μm particle size, L × I.D. 100 × 4.6 mm, Phenomenex (Torrance, CA, USA) with electrochemical detection (750 mV, DECADE II™ Electrochemical Detector; Antec (Hoorn, The Netherlands) and a glassy carbon flow cell (VT-03 cell 3 mm GC sb; Antec) using a Shimadzu chromatograph (Shimadzu Corporation, Kyoto, Japan). One liter of mobile phase (pH = 3.2) contained 13.06 g of KH_2_PO_4_, 200 μL of 0.5 M Na_2_EDTA, 300 mg of 1-octanesulfonic acid sodium salt (Sigma, St. Louis, MO, USA), 940 μL of H_3_PO_4_ and 130 μL of methanol (Vektor Ltd., Novosibirsk, Russia) [24]. The standard mixes containing 0.5, 1 and 2 ng of 5-HT and 5-HIAA were repeatedly assayed throughout the entire procedure and used to plot the calibration curves for each substance. The areas of peaks were estimated using LabSolution LG/GC software version 5.54 (Shimadzu Corporation, Kyoto, Japan) and calibrated against the calibrated curves for corresponding standards. The 5-HT and 5-HIAA contents were expressed in ng/mg protein assayed by the Bradford method and described elsewhere [24]. All data were means of three replications.

#### 2.5.3. Assay of the TPH Activity

An aliquot of 15 μL of pure supernatant (Section 2.5.1) was incubated for 15 min at 37 °C with L-tryptophan (Sigma, USA) (0.4 mM), cofactor 6-methyl-5,6,7,8-tetrahydropteridine (Sigma, USA) (0.3 mM), decarboxylase inhibitor m-hydroxybenzylhydrazine (Sigma, USA) (0.3 mM), catalase (Sigma, USA) (5 U) and 1 mM dithiothreitol in the final volume of 25 μL. The reaction was stopped with 75 μL 0.6 M HClO_4_ and centrifuged for 15 min at 12,700× *g* rpm. The clear supernatant was diluted by twofold with ultra-pure water and the 5-HTP concentration was determined in the diluted supernatant using high-performance liquid chromatography (see Section 2.5.2) using standards of 25, 50 and 100 pmoles of 5-HTP (Sigma, USA). The TPH2 activity was expressed as pmoles 5-HTP formed per minute per mg of protein measured according to the Bradford method [24].

#### 2.5.4. Assay of TPH2 K_M_ and V_max_ Values

For the assay of K_M_ and V_max_ values for BH_4_, an aliquot of 10 μL of the pooled supernatant was incubated for 15 min at 37 °C with 0.3 mM of L-tryptophan and 1.2, 0.6, 0.3, 0.15, 0.075, 0.0375 or 0.01875 mM of BH_4_. For the assay of K_M_ and V_max_ values for L-tryptophan, an aliquot of 10 μL of the pooled supernatant was incubated for 15 min at 37 °C with 0.3 mM of BH_4_ and 0.4, 0.2, 0.1, 0.05, 0.025 or 0.0125 mM of L-tryptophan. The TPH2 activity was assayed as described earlier (Section 2.5.3) and expressed in pmoles 5-HTP formed per minute per mg of protein measured according to the Bradford method [24].

#### 2.5.5. Assay of TPH2 Thermal Stability

Aliquots of 15 μL of pooled supernatant (Section 2.5.1) were heated for 2 min at 48–64 °C (2 °C interval) and put in ice, while the control tubes were not heated. Then, the TPH2 activity was assayed as described earlier (Section 2.5.3) [28]. Two groups of 5–6 thermal curves for TPH2 from B6-1473C and B6-1473G were plotted.

#### 2.5.6. TPH2 Protein Quantification with Western Blot Analysis

The TPH2 levels were determined by means of Western blot analysis as described earlier [29]. For TPH2 protein detection (at the 56 kDa level), a polyclonal rabbit antibody was used (1:1000 dilution, cat. # ab184505, Abcam, Cambridge, UK). The GAPDH protein level was used as internal control, detected with a polyclonal rabbit antibody (1:2000, ab9485, Abcam) at 37 kDa. Original figures can be found in Appendix A.

#### 2.5.7. mRNA Level Assay by qPCR

Total mRNA was extracted from the mix of homogenate with Trizol reagent (see Section 2.5.1) according to the manufacturer’s protocol, treated with RNAase free DNAase (Promega, Medisson, WI, USA) according to the manufacturer’s protocol, its concentration was assayed with Nanodrop 2000 (Thermo Fisher Inc., Waltham, MA, USA) and it was diluted to the final concentration of 125 ng/µL. The cDNA was synthesized using a random hexanucleotide primer and R01 Kit according to the manufacturer’s protocol (Biolabmix, Novosibirsk, Russia). The mRNA level of target genes was assayed by qPCR using the set of selective primers (Table 1) and R401 Kit (Sintol, Moscow, Russia) according to the manufacturer’s protocol (95 °C 5 min; (95 °C, 15 s; annealing temperature, 60 s; 82 °C, 2 s; fluorescence registration) × 40 cycles). The threshold cycles were calibrated with the external standards containing 25, 50, 100, 200, 400, 800, 1600, 3200 and 6400 copies of genomic DNA extracted from C57BL/6 mouse liver. The gene expression was presented as a relative number of cDNA copies calculated on 100 copies of *Polr2a* cDNA as an internal standard [24].

### 2.6. Statistics

#### 2.6.1. Calculation of K_M_ and V_max_ Values

The K_M_ and V_max_ values and their errors were calculated by the Jehannes and Lumry algorithm [30] and compared by applying Student’s *t* test.

#### 2.6.2. Analysis of the Thermal Denaturation Curves

For T_50_ calculation, the thermal denaturation curves in the coordinates T and (Vc − Vt)/Vc were used (T—heating temperature (48–64 °C), Vt—the TPH2 activity after heating at temperature t, Vc—the TPH2 activity in the control sample). The linear part of each curve was approximated by Equation (1) T = b × (Vc − Vt)/Vc + a,(1)
and the b and a coefficients were calculated. Using these coefficients, the T_50_ value was calculated by replacing (Vc − Vt)/Vc by 0.5 in Equation (1) (T_50_ = b × 0.5 + a) [28].

For calculation of the free Gibbs energy of denaturation (ΔG), the equation ΔG = ΔH − ΔS × T was used (T—Kelvin temperature °K, ΔH and ΔS—enthalpy and entropy of denaturation). When T = T_50_, ΔG = 0 and ΔH = ΔS/T_50_. The slope coefficient (d) of Equation (2)(Vc − Vt)/Vc = d × T + c(2)
curves equally to ΔH/R × (T_50_)^2^ [31] (R—gas constant, 1.987 cal/grad/mole). The thermal denaturation curves in the coordinates (Vc − Vt)/Vc and T (T—Kelvin temperature of heating, 321–337 °K) were plotted. The linear part of each curve was approximated by Equation (2) and the d and c coefficients were calculated. Using the d and T_50_ values, ΔH = d × R × (T_50_)^2^ and ΔS = ΔH/T_50_ were evaluated. Using these ΔH and ΔS values, the standard ΔG value (at 25 °C) was calculated according to the equation ΔG = ΔH − ΔS × 298.15.

#### 2.6.3. Statistical Tests

Behavioral traits in the open field, tail suspension, olfactory preference tests and biochemical data were presented as the mean ± SEM and analyzed using one-way or two-way ANOVA. The distance travelled and the sleep duration in the home cage test were analyzed via two-way repeated measures ANOVA with “Day” as the within variable. The number and duration of male urine sniffing by females were analyzed via one-way repeated measures ANOVA with the male’s genotype as the within variable. Post hoc analyses were carried out using Fisher’s LSD multiple comparison test when appropriate. The dominance indexes were analyzed by the Mann–Whitney U test. The numbers of genotypes were compared by the χ^2^ test. Statistical significance was set at *p* < 0.05.

## 3. Results

### 3.1. The K_M_ and V_max_ Values for BH_4_ and L-Tryptophan for TPH2 from the Midbrains of B6-1473C and B6-1473G Males

The C1473G mutation markedly decreases V_max_ values both for BH_4_ and L-tryptophan (Table 2, Figure 5). At the same time, this substitution does not alter K_M_ values for BH_4_ and even markedly decreases K_M_ values for L-tryptophan (Table 2, Figure 5).

### 3.2. Tph2 Gene Expression and TPH2 Protein Density and Activity in the Midbrains of B6-1473C and B6-1473G Adult Males

As expected, the TPH2 activity in the midbrains of B6-1473C males was markedly higher than that in the midbrains of B6-1473G males (F_1,14_ = 6.87, *p* = 0.02, Figure 6A). No difference in the *Tph2* gene mRNA level in the midbrains of B6-1473C and B6-1473G males was observed (F_1,13_ = 1.64, *p* = 0.22, Figure 6B). At the same time, the TPH2 protein level was higher in the midbrains of B6-1473C males compared to that in the midbrains of B6-1473G males (F_1,15_ = 8.84, *p* = 0.009, Figure 6C).

### 3.3. Effect of the C1473G Mutation on the TPH2 Molecule Thermal Stability In Vitro

The C1473G mutation (P447R) substitution markedly decreases both the T_50_ (F_1,14_ = 17.84, *p* < 0.001, Figure 7A) and ΔG (F_1,13_ = 46.67, *p* < 0.001, Figure 7B) of TPH2 thermal denaturation.

### 3.4. Tube Competition of B6-1473C and B6-1473G Adult Males

No difference in the dominance indexes in the tube competition test between adult B6-1473C and B6-1473G males was observed (Z = −0.049, n = 9, *p* > 0.05, Figure 8).

### 3.5. Olfactory Preference for B6-1473C and B6-1473G Adult Male Urine by Receptive B6-1473C and B6-1473G Adult Females

One-way ANOVA for repeated measurements did not show any effect of the “Female genotype” (number of sniffings, F_1,21_ < 1; sniffing time, F_1,21_ < 1) and “Male’s urine genotype” (number of sniffings, F_1,21_ =1.68, *p* = 0.21; sniffing time, F_1,21_ < 1) factors. At the same time, a marked effect of these factors’ interaction on the number of sniffings (F_1,21_ =4.33, *p* = 0.049), but not sniffing time (F_1,21_ < 1), was revealed. B6-1473G females more frequently sniffed the urine of B6-1473G males compared to that of B6-1473C males (Figure 9).

### 3.6. Distribution of Homo- and Heterozygote Litters in B6-1473C and B6-1473G Adult Females Put into Cage with a Pair of B6-1473C and B6-1473G Adult Males

No differences in the numbers of homo- and heterozygous litters in B6-1473C (homozygous 9, heterozygous 10) and B6-1473G (homozygous 13, heterozygous 12) adult females were detected (χ^2^ = 0.09, df = 1, *p* = 0.76, Figure 10).

### 3.7. Body Masses, 5-HT and 5-HIAA Levels, TPH2 Activity, and Tph1 and Tph2 Gene mRNA Levels in the Brains of 19-Day-Old Embryos of the B6-1473CC, B6-1473CG and B6-1473GG Genotypes

Among 48 embryos from five B6-1473CG females mated by B6-1473CG males, the distribution of genotypes was the following: 10 B6-1473CC, 24 B6-1473CG and 14 B6-1473GG. These numbers did not differ from the expected 1:2:1 segregation ratio (χ^2^ = 0.67, df = 2, *p* = 0.28).

No differences in the body masses of 19-day-old embryos of the B6-1473CC, B6-1473CG and B6-1473GG genotypes were observed (F_2,41_ = 1.64, *p* = 0.21, Figure 11).

No differences in the 5-HT levels in the whole brains of 19-day-old embryos of the B6-1473CC, B6-1473CG and B6-1473GG genotypes were shown (F_2,36_ = 0.7, *p* = 0.5, Figure 12A). At the same time, marked differences in the 5-HIAA levels (F_2,36_ = 18.41, *p* < 0.001, Figure 12B) and the 5-HIAA/5-HT ratios (F_2,36_ = 9.29, *p* < 0.001, Figure 12C) in the whole brains of 19-day-old embryos of the B6-1473CC, B6-1473CG and B6-1473GG genotypes were revealed. These two traits were higher in the brains of B6-1473CG and B6-1473GG embryos compared to B6-1473CC embryos (Figure 12B,C).

No differences in the TPH2 activity (F_2,36_ = 1.87, *p* = 0.17, Figure 13A) and the *Tph2* (F_2,36_ = 0.92, *p* = 0.41, Figure 13B) and *Tph1* (F_2,36_ = 1.08, *p* = 0.35, Figure 13C) gene mRNA levels in the whole brains of 19-day-old embryos of the B6-1473CC, B6-1473CG and B6-1473GG genotypes were shown. It is worth noting that the mRNA levels of the *Tph1* gene were very high: this gene expression was about 20% of the *Tph2* gene expression.

### 3.8. Hind Legs Clasping in Juvenile and Adult B6-1473CC and B6-1473GG Males

The “Genotype” and “Age” factors (but not these factors’ interaction) had marked effects on the frequency and accumulated time of hind legs clasping (Table 3). High frequency and accumulating time of hind legs clasping were observed only in juvenile (3 weeks old) B6-1473G males, but not in juvenile B6-1473C males and adult (12 weeks old) B6-1473G males (Figure 14).

### 3.9. Prolonged Effects of LPS and Saline Administration on Survival, Home Cage Activity, Open Field Behavior and 5-HT System Characteristics in the Midbrains of Juvenile B6-1473C and B6-1473G Males

Body masses of juvenile (3 weeks old) B6-1473G males were lower than those of juvenile B6-1473C males (F_1,46_ = 5.8, *p* = 0.02, Figure 15).

Five of twelve (41.6%) juvenile B6-1473G males died during the 72 h after a single LPS (2 mg/kg) administration, while all eight juvenile B6-1473C males were alive after LPS treatment (χ^2^ = 4.44, df = 1, *p* = 0.035). All juvenile B6-1473G (n = 8) and B6-1473C (n = 8) males were alive after a single saline administration. Below, the data on the animals that survived after LPS and saline administration are presented.

The influence of the “Genotype” and “Day” factors on the variability of the distance travelled and the accumulated sleep duration per day in the home cage (Phenomaster) was revealed (Table 4). As expected, during the first 24 h after LPS administration, animals of both genotypes moved less and slept longer compared to the control ones. Over the next 4 days, these parameters gradually normalized (Figure 16). B6-1473C mice generally moved more and slept less than B6-1473G animals (Figure 16). No influence of the “Treatment” factor and of the interaction of the “Genotype” × “Treatment” factors on these parameters was revealed. However, an influence of the “Day” × “Treatment” interaction on sleep duration was found (Table 4).

Five days after the administration of saline or LPS, no effects of the factors “Genotype” (F_1,25_ = 0.56, *p* = 0.46) and “Treatment” (F_1,25_ = 0.0, *p* = 1.0), or of their interaction (F_1,25_ = 0.37, *p* = 0.55), on the time spent in the center were found in the open field test. Mice of all groups avoided the central part of the arena to the same extent (Figure 17). At the same time, a significant effect of the “Genotype” factor (F_1,25_ = 14.21, *p* < 0.001), but not the “Treatment” factor (F_1,25_ = 1.78, *p* = 0.19) or the factors’ interaction (F_1,25_ < 0.65, *p* = 0.43), on the distance travelled was found (Figure 17). The saline-treated B6-1473G mice ran longer distances compared to the saline-treated B6-1473C mice (Figure 17).

Two-way ANOVA revealed a significant effect of the “Treatment” factor (F_1,28_ = 24.78, *p* < 0.001), but not the “Genotype” factor (F_1,28_ = 1.85, *p* = 0.18) and the factors’ interaction (F_1,28_ = 0.49, *p* = 0.49), on the spleen/body mass ratios. This index did not differ in the saline-treated males of both genotypes, while it increased 5 days after a single LPS administration in mice of both genotypes (Figure 18A). Significant effects of the “Genotype” (F_1,28_ = 8.81, *p* = 0.006) and “Treatment” (F_1,28_ = 88.03, *p* < 0.001) factors, but not the factors’ interaction (F_1,28_ = 2.9, *p* = 0.1), on the thymus/body mass ratio were shown. This index was low in the saline-treated B6-1473G mice compared to the saline-treated B6-1473C mice (Figure 18B). Five days after the administration of LPS, the thymus/body mass ratio decreased in males of both genotypes (Figure 18B).

Five days after the saline and LPS injections, no differences were found in the 5-HT (F_3,28_ = 2.2, *p* = 0.11, Figure 19A) and 5-HIAA (F_3,28_ = 1.72, *p* = 0.18, Figure 19B) levels and the 5-HIAA/5-HT ratios (F_3,28_ = 1.52, *p* = 0.23, Figure 19C) in the midbrains of the survived juvenile males of the four groups. A high effect of the “Genotype” factor (F_1,28_ = 11.2, *p* = 0.002), but not of the “Treatment” factor (F_1,28_ = 0.02, *p* = 0.89) and the factors’ interaction, on the TPH2 activity in the midbrain was revealed. As expected, the enzyme activity in this structure was higher in B6-1473C mice compared to B6-1473G animals (Figure 19D). The enzyme activity in the midbrain did not change 5 days after the administration of LPS compared to that after the administration of saline in mice of both genotypes (Figure 19D).

Five days after the saline and LPS injections, no differences in the mRNA levels of *Tnf* (F_3,28_ = 1.32, *p* = 0.29, Figure 20B), *Tph2* (F_3,28_ = 0.23, *p* = 0.87, Figure 20C), *Slc6a4* (F_3,28_ = 0.23, *p* = 0.87, Figure 20D), *Htr1a* (F_3,28_ = 2.33, *p* = 0.10, Figure 20E) and *Htr2a* (F_3,28_ = 0.18, *p* = 0.91, Figure 20F) genes in the midbrain were detected in the survived juvenile males of four experimental groups. At the same time, a significant effect of the “Treatment” factor (F_3,28_ = 14.19, *p* < 0.001), but not of the “Genotype” factor (F_3,28_ = 2.77, *p* = 0.11) and the factors’ interaction (F_3,28_ = 0.0, *p* = 0.99), on the mRNA level of the *Il1b* gene in the midbrain was detected. A high level of expression of this gene was observed in this structure in mice of both genotypes 5 days after LPS administration (Figure 20A).

## 4. Discussion

Twenty years ago, M. Caron and his colleagues discovered the C1473G polymorphism in the mouse *Tph2* gene that caused the P447R substitution in the mouse TPH2 molecule [14]. Later, an association [16] of this polymorphism with the previously discovered [15] genetically determined variability in the enzyme activity in the brain of laboratory mice was shown. The 1473G allele reduces enzyme activity, but the molecular mechanism of this reduction remains obscure.

Although a natural selection against the 1473G allele in wild mouse populations was shown [20], the physiological and behavioral mechanisms that cause the mutant allele elimination are still unknown. Indeed, three independent groups of researchers using B6-1473C and B6-1473G congenic lines, differing only in the 1473C and 1473G alleles and enzyme activity, did not reveal any visible and reproducible effects of the mutant allele (1473G) on the behavior (in a standard battery of tests) and physiological functions of adult mice [17,18,19].

Therefore, the main aims of the present study were the explanation of (1) the molecular mechanism decreasing the mutant TPH2 activity and (2) some physiological and behavioral mechanisms altered by this mutation.

### 4.1. Molecular Mechanism Decreasing the Mutant TPH2 Activity

From general considerations, the C1473G mutation can reduce the TPH2 activity by the following hypothetical molecular mechanisms: (1) decrease in the enzyme affinity to L-tryptophan and/or BH_4_, (2) decrease in the *Tph2* gene expression and/or mRNA stability and (3) decrease in TPH2 protein translation/or stability.

In order to test the first mechanism, the K_M_ values for L-tryptophan and BH_4_ for the wild-type and mutant enzymes negatively correlating with the enzyme affinity (the higher the K_M,_ the lower the affinity is) were defined. Neither of these enzyme forms differed in their K_M_ for BH_4_. Moreover, the K_M_ for L-tryptophan for the mutant TPH2 was even lower (and therefore its affinity was higher) than that for the wild-type enzyme. These results agree with those achieved using recombinant wild-type and mutant proteins [32]. Therefore, a possible alteration in the enzyme’s affinity for substrate/cofactor cannot explain the observed decrease in the mutant enzyme activity.

The present results and those of Sakowsky [32] indicate that the mutation causes a significant decrease in the V_max_. A possible cause of the observed decrease in the mutant TPH2 V_max_ can be a reduction in the *Tph2* gene mRNA or the TPH2 protein expression/stability. The *Tph2* gene expression is specific for the 5-HT neuron cell bodies located in the midbrain [33,34]. In the present study, no alteration in the *Tph2* gene mRNA levels in the midbrains of B6-1473C and B6-1473G mice was observed. Earlier, no effect of the C1473G mutation on the *Tph2* gene mRNA levels in the midbrains of B6-1473C and B6-1473G mice was observed [35,36]. Therefore, the C1473G mutation does not seem to alter the *Tph2* gene mRNA expression or stability.

At the same time, the first time Western blot analysis revealed a marked decrease in the TPH2 protein levels in the midbrains of B6-1473G mice compared to those of B6-1473C mice. This observed decrease can result from an alteration in the mutant TPH2 protein translation or stability. In the present study, an effect of P447R substitution on TPH2 stability was evaluated.

The functional activity of a protein is determined by its 3D structure, which in turn is supported by electrostatic bonds between amino acid residues. The destruction of these bonds leads to the formation of a chaotic protein structure and partial or complete loss of its activity. This process is known as protein denaturation. Substitution of an amino acid can weaken these bonds and facilitate protein denaturation. Denaturation is a statistical process that occurs continuously at body temperature when the protein molecule is bombarded with water molecules and ions. Many enzymes have a short lifespan (several days). In experimental studies of denaturation, higher temperatures are used to speed up this process.

Protein stability is estimated by the Gibbs free energy of its denaturation (ΔG), i.e., the energy that must be imparted to the protein to destroy the bonds that stabilize its structure [31]. The higher the denaturation energy, the more stable the protein is. Protein stability is usually assayed by its T_50_ value—the temperature at which 50% of the protein is denatured [37,38,39,40,41]. In the present study, we calculated the ΔG of denaturation for the first time. The analysis of thermal denaturation curves for TPH2 extracted from midbrains of B6-1473C and B6-1473G mice showed that the P447R substitution reduces the T_50_ and ΔG of the mutant TPH2 molecule. Thus, the results of this study provided experimental evidence that the P447R substitution reduces the stability of the mutant TPH2, which reduces its lifetime, the density of molecules in the cell and, ultimately, the enzyme activity.

### 4.2. Physiological and Behavioral Traits Altered by the C1473G Mutation

Adult B6-1473C and B6-1473G males maintained in an SPF vivarium do not differ in their body mass [19], aggressive behavior [35] and numerous standard physiological and behavioral characteristics [19]. It is important to note that the standard battery of tests used in the Koshimizu [19] study was developed for pharmacological research and apparently does not reflect the factors affecting the ability of mice to survive in natural conditions. That is why as the first step of the present study the effects of the C1473G mutation on hierarchical state, competition and attraction for a sexual partner were investigated. The 1473G allele does not seem to alter the intraspecific hierarchy and ability to mate females in adult males. Moreover, wild-type and mutant females do not show any olfactory preference for wild-type or mutant males. These findings, together with the published data [19], indicate that the 1473G allele does not seem to alter the adaptive behavior of adult males.

As the second step of the study, the effects of the C1473G mutation on body mass and the brain 5-HT system of 19-day-old mouse embryos were studied. No effect of the C > G substitution on the body mass and survival of mouse embryos was observed. Unexpectedly, the TPH2 activity in the whole brains of embryos with 1473CC, 1473CG and 1473GG genotypes did not differ. Moreover, a relatively high expression of the *Tph1* gene in the brains of mouse embryos was revealed. This finding is very interesting since no expression of the *Tph1* gene in adult mice brains was shown [3,4,6,42,43,44]. This high TPH1 expression seems to smooth out an expected difference in the total TPH activity in the brains of B6-1473C and B6-1473G embryos. The level of 5-HIAA in the brains of mutant and heterozygous embryos is surprisingly higher compared to the wild-type embryos. An explanation of this increase of 5-HT turnover in the brains of mutant embryos is a subject for further study. These results indicate that the 1473G allele produces a small if any effect on mouse embryos.

In the present study and a previous [24] study, a slight but statistically significant decrease in the body mass of young and adult mutant mice compared to wild-type animals was shown. This decrease in body mass can be considered as a factor indirectly reducing concurrent and reproductive success of adult mutant males, resulting in the 1473G allele elimination in natural mouse populations.

The 1473G allele produces a more considerable negative effect on juvenile mice when they leave their home nest and start their independent life. First of all, this allele causes pronounced hind legs dystonia in juvenile B6-1473G mice. It may be assumed that due to this dystonia mutant, juvenile mice less effectively escape from predators compared to wild-type mice and it is another factor decreasing the mutant allele rate in natural mouse populations.

Besides predators, bacterial, viral and parasitic infections are another key environmental factor reducing the progressive rise of natural populations [45]. In the present study, we showed that a single LPS administration killed more than 40% of young B6-1473G mice without any lethal effect on young B6-1473C mice.

A single LPS administration causes a prolonged activation of the innate immune system and sickness syndrome including a dramatic decrease in general activity [46]. In this paper, we showed an increased *Il1b* gene expression in the midbrain as well as an increased spleen mass and a decreased thymus mass on the 6th day after LPS administration. At the same time, we revealed that locomotor activity normalized on the 6th day after LPS administration.

The brain 5-HT system seems to be involved in the regulation of innate immunity: LPS increases 5-HT metabolism in the brain [47]. Recently, we showed that the LPS-induced activation of 5-HT metabolism in the brains of adult mice was continuing for at least for 24 h after a single toxin administration [48]. The brain 5-HT system was normalizing for several days after LPS administration, i.e., we did not observe alterations in the brain 5-HT metabolism and the expression of *Tph2, Slc6a4, Htr1a,* and *Htr2a* genes in the midbrains of the B6-1473G and B6-1473C mice on the 6th day after administration. Moreover, the C1473G polymorphism seems to be crucial for the LPS-induced activation of the brain 5-HT system: this activation was blunted in the brains of B6-1473G mice compared to B6-1473C mice [48]. It can be assumed that a high TPH2 activity can restore the loss of 5-HT caused by its LPS-induced metabolism activation, while a low TPH2 activity of the mutant enzyme does not seem to be sufficient to restore the LPS-induced loss of 5-HT. This decreases the protected function of 5-HT and results in death of juvenile mutant mice.

In the present study, we showed that young mutant mice were more vulnerable to innate immune system stimulation. A single LPS administration harmless for young wild-type mice can kill about 40% of young mutant mice. It can be hypothesized that young mutant mice more frequently die due to infections compared to young wild-type mice. This high vulnerability of mutant mice to innate immunity stimulation seems to be the third factor that eliminates the 1473G allele in natural mouse populations.

## 5. Conclusions

The C1473G polymorphism in the *Tph2* gene that defined the TPH2 activity in mouse brain was discovered by M. Caron and his colleagues. In the present study, using in vitro biochemical and in vivo physiological and behavioral techniques, we traced the sequence of molecular and physiological events from the C > G substitution to the elimination of mutant mice in natural populations for the first time.

In vitro, a decrease in the Gibbs free energy of denaturation of the mutant TPH2 compared to the wild-type TPH2 was shown. This decrease in free energy of denaturation indicates that the C > G (or *p* > R) substitution reduces the stability, living time and activity of mutant TPH2.

The 1473G allele does not seem to directly alter the intraspecific hierarchy and the ability to mate females in adult males. Moreover, females do not show any olfactory preference for wild-type or mutant males. These findings, together with the published data, indicate that the 1473G allele does not seem to alter the adaptive behavior of adult males.

At the same time, the 1473G allele seems to slightly, but significantly, reduce the body mass of mutant males. This can be considered as a factor indirectly reducing concurrent and reproductive success of adult mutant males, resulting in the 1473G allele elimination in natural mouse populations.

We showed that the C > G substitution did not alter the body mass and survival of mouse embryos. Unexpectedly, a relatively high expression of the *Tph1* gene in the brain of mouse embryos was revealed. This high TPH1 expression seems to smooth out an expected difference in the total TPH activity in the brains of B6-1473C and B6-1473G embryos. These results indicate that the 1473G allele produces a small effect on mouse embryos.

The 1473G allele produces a more considerable negative effect on the survival of young mice when they leave their home nest and start their independent life. First of all, this allele produces pronounced hind legs dystonia in young B6-1473G mice. Due to this dystonia, young mice cannot effectively escape from predators, and it is another factor decreasing the mutant allele rate in the natural mouse population.

We showed that young mutant mice were more vulnerable to innate immune system stimulation. A single LPS administration, harmless for young wild-type mice, can kill about 40% of young mutant mice. It can be hypothesized that young mutant mice more frequently die due to infections and parasites compared to young wild-type mice. This high vulnerability of mutant mice to innate immunity stimulation seems to be the third factor that eliminates the 1473G allele in natural mouse populations.

## Figures and Tables

**Figure 1 biomolecules-15-00461-f001:**
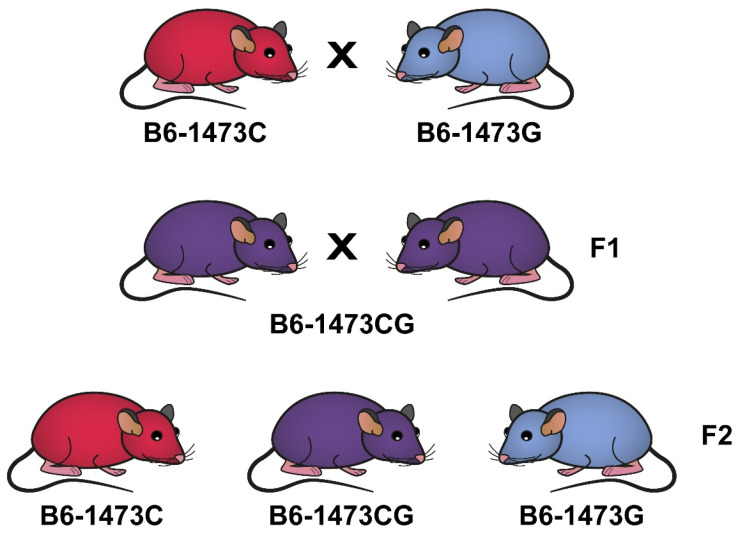
Breeding B6-1473C and B6-1473G congenic mice homozygous for 1473C and 1473G alleles, respectively.

**Figure 2 biomolecules-15-00461-f002:**
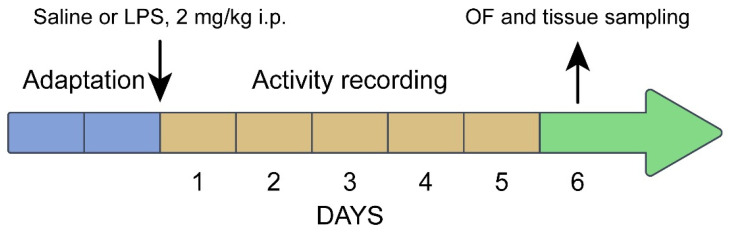
Timeline of Experiment 7.

**Figure 3 biomolecules-15-00461-f003:**
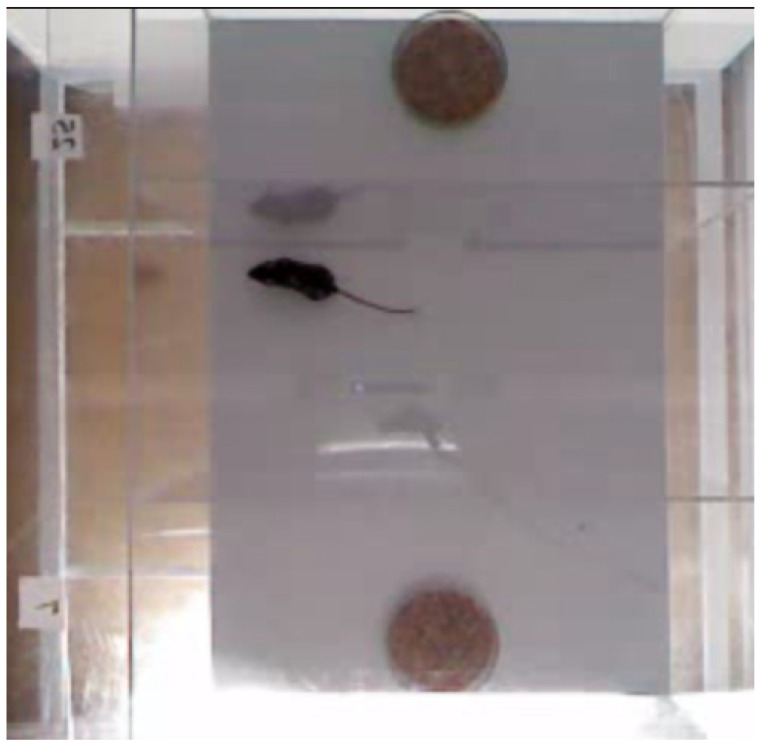
Olfactory preference test. Female is in the central compartment, while two Petri dishes filled with beddings from B6-1473C and B6-1473G males are in two side compartments. Photo made by EthStudio software.

**Figure 4 biomolecules-15-00461-f004:**
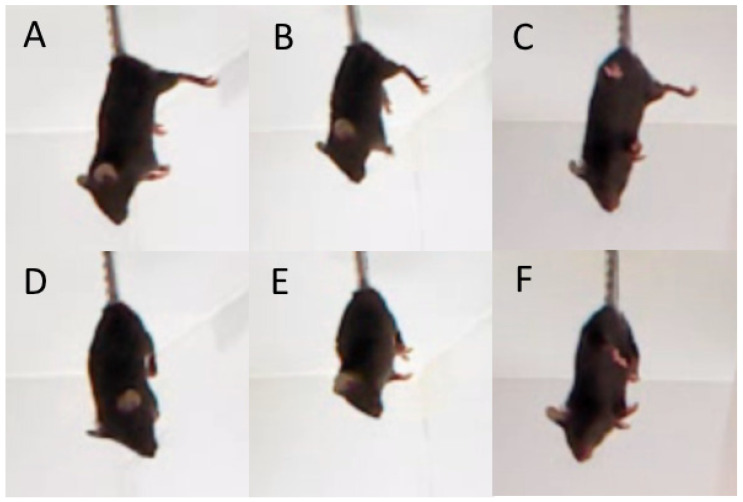
Hind legs position in the tail suspension test. (**A**–**C**) normal hind legs position, (**D**–**F**) hind legs clasping. Photo made by EthStudio software.

**Figure 5 biomolecules-15-00461-f005:**
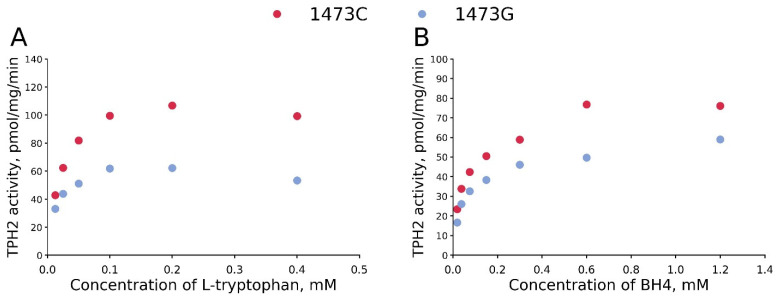
Dependence of the activity of 1473C and 1473G forms of TPH2 on L-tryptophan (**A**) and BH_4_ (**B**) concentrations. TPH2 extracted from midbrains of B6-1473C and B6-1473G males. Each point is the mean of three measurements.

**Figure 6 biomolecules-15-00461-f006:**
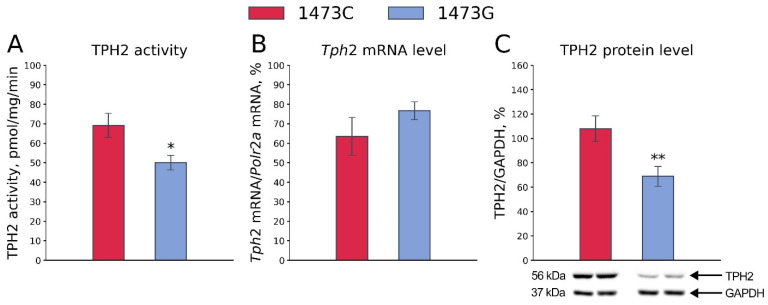
TPH2 activity (**A**), *Tph2* gene mRNA level (**B**) and TPH2 protein level (**C**) in the midbrains of adult B6-1473C and B6-1473G males. Each value is mean ± SEM for 7–9 animals. * *p* < 0.05, ** *p* < 0.01 vs. B6-1473C.

**Figure 7 biomolecules-15-00461-f007:**
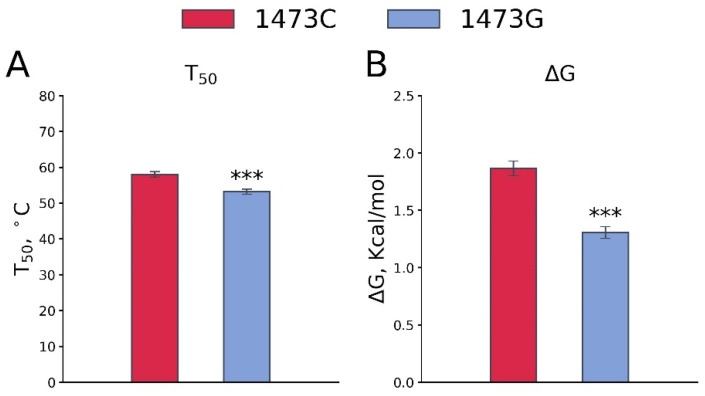
T_50_ (**A**) and ΔG (**B**) values of the thermal stability of TPH2 from midbrains of B6-1473C and B6-1473G males. Each value is mean ± SEM for 8 thermal denaturation curves. *** *p* < 0.001 vs. B6-1473C.

**Figure 8 biomolecules-15-00461-f008:**
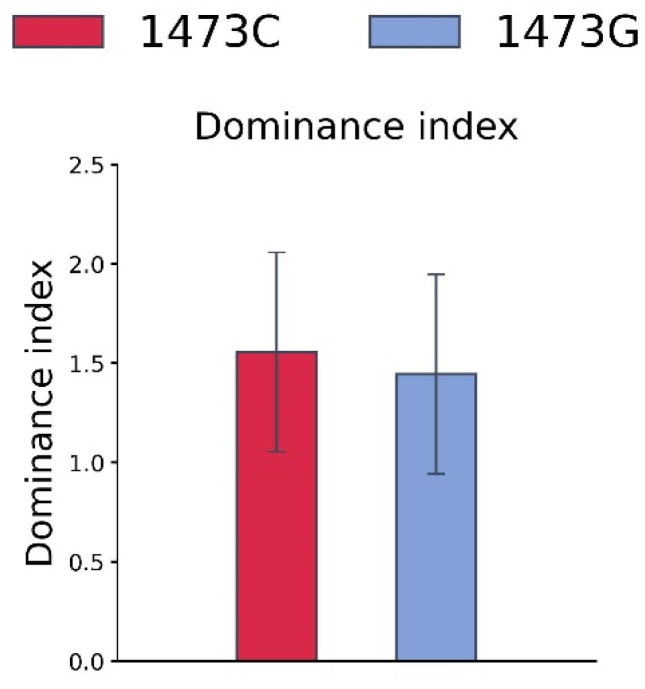
Dominance index for B6-1473C and B6-1473G adult males in the tube competition test. Each value is mean ± SEM for nine observations.

**Figure 9 biomolecules-15-00461-f009:**
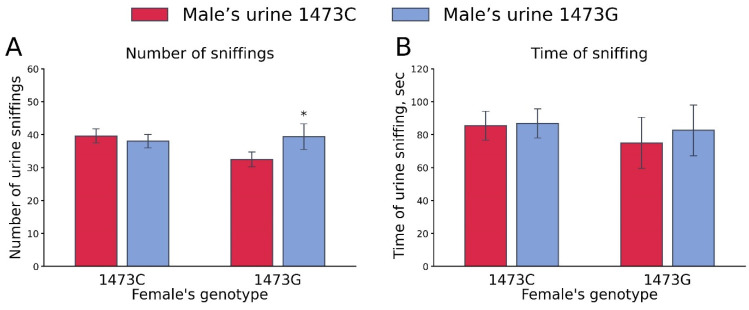
Number of sniffs (**A**) and sniffing time (**B**) of urine of B6-1473C and B6-1473G males by B6-1473C (n = 15) and B6-1473G (n = 8) females. * *p* < 0.05 vs. urine of B6-1473C males.

**Figure 10 biomolecules-15-00461-f010:**
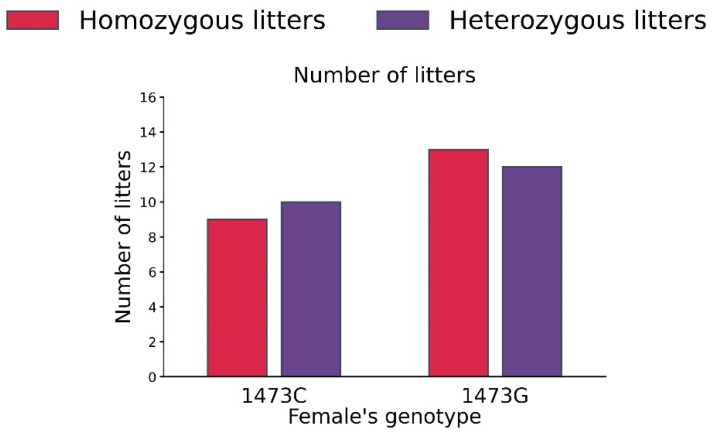
Numbers of homo- and heterozygous litters in B6-1473C (n = 19) and B6-1473G (n = 25) adult females put into cage with pair of B6-1473C and B6-1473G adult males.

**Figure 11 biomolecules-15-00461-f011:**
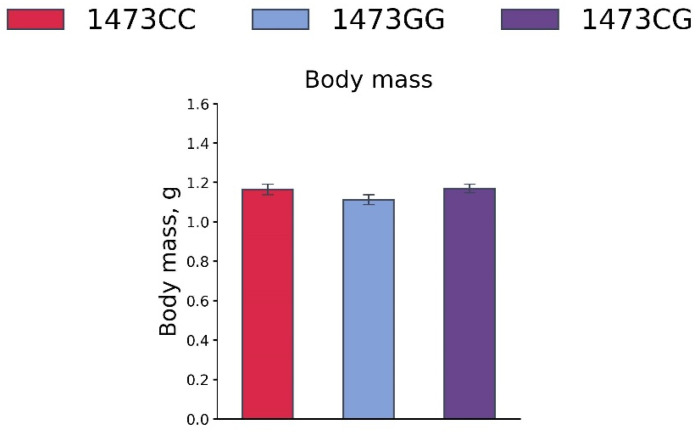
Body masses of 19-day-old embryos of the B6-1473CC (n = 10), B6-1473CG (n = 20) and B6-1473GG (n = 14) genotypes.

**Figure 12 biomolecules-15-00461-f012:**
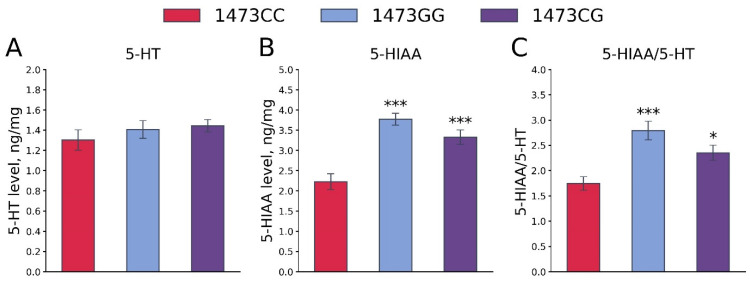
5-HT (**A**) and 5-HIAA (**B**) levels and 5-HIAA/5-HT ratios (**C**) in the whole brains of 19-day-old embryos of the B6-1473CC (n = 10), B6-1473CG (n = 15) and B6-1473GG (n = 14) genotypes. * *p* < 0.05, *** *p* < 0.001 vs. B6-1473CC.

**Figure 13 biomolecules-15-00461-f013:**
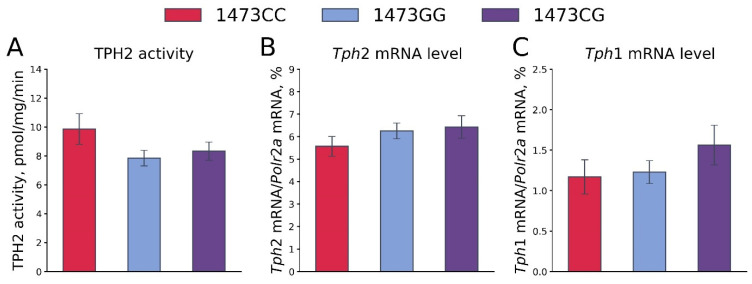
TPH2 activity (**A**) and *Tph2* (**B**) and *Tph1* (**C**) gene mRNA levels in the whole brains of 19-day-old embryos of the B6-1473CC (n = 10), B6-1473CG (n = 15) and B6-1473GG (n = 14) genotypes.

**Figure 14 biomolecules-15-00461-f014:**
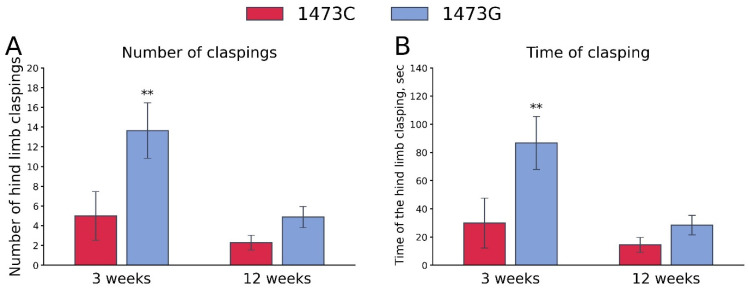
Number (**A**) and duration (**B**) of hind paws clasping in juvenile (3 weeks old) and adult (12 weeks old) B6-1473C (juvenile, n = 16; adult, n = 21) and B6-1473G males (juvenile, n = 28; adult, n = 24). ** *p* < 0.01 vs. B6-1473G of the same age.

**Figure 15 biomolecules-15-00461-f015:**
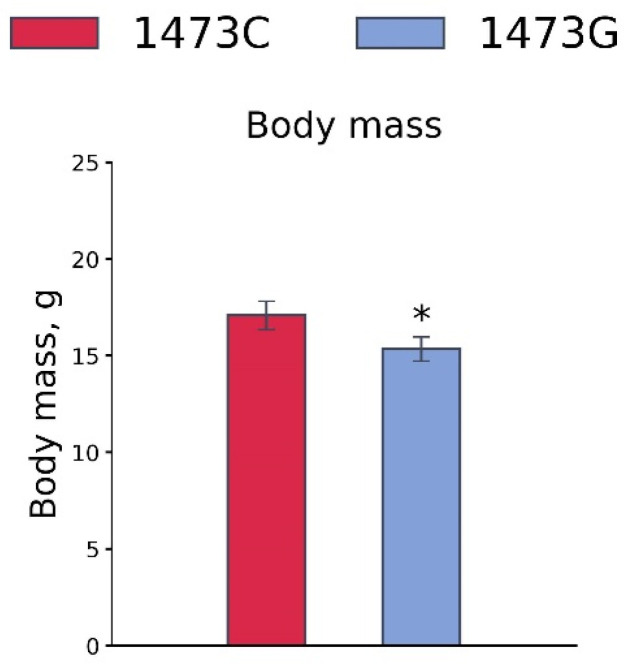
Body masses of 3-week-old B6-1473C (n = 24) and B6-1473G (n = 24) males. * *p* < 0.05 vs. B6-1473C.

**Figure 16 biomolecules-15-00461-f016:**
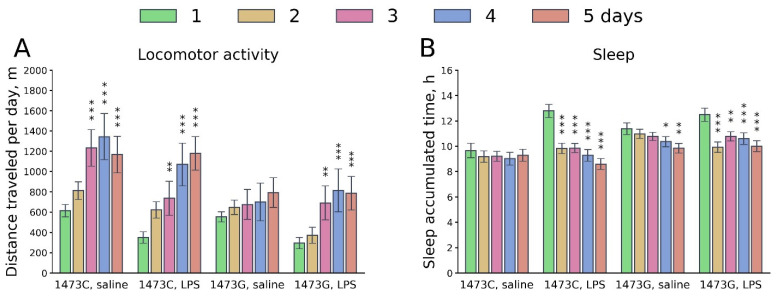
Dynamics of day distance travelled (m per day, (**A**)) and accumulated sleep time (hours per day, (**B**)) in the home cages of survived juvenile (3 weeks) B6-1473C (saline-treated, n = 7; LPS-treated, n = 7) and B6-1473G males (saline-treated, n = 9; LPS-treated, n = 7) after a single intraperitoneal injection of saline or 2 mg/kg LPS. * *p* < 0.05, ** *p* < 0.01, *** *p* < 0.001 vs. animals of the same group on the first day after injection.

**Figure 17 biomolecules-15-00461-f017:**
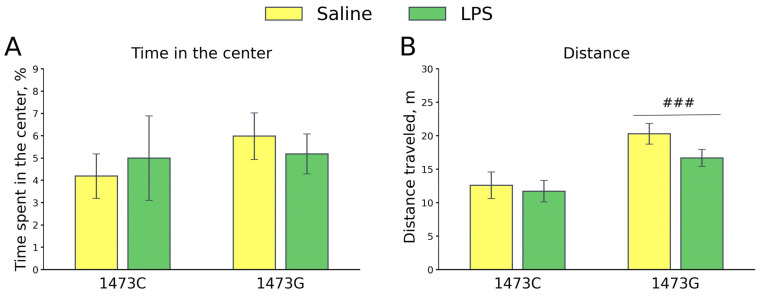
Time spent in the center (%, (**A**)) and travelled distance (m, (**B**)) in the open field test of survived juvenile B6-1473C (saline-treated, n = 7; LPS-treated, n = 7) and B6-1473G (saline-treated, n = 10; LPS-treated, n = 6) males five days after single ip saline or LPS (2 mg/kg) administration. ### *p* < 0.001 vs. B6-1473C mice.

**Figure 18 biomolecules-15-00461-f018:**
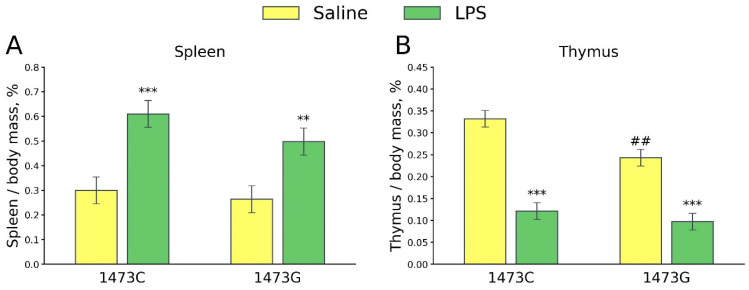
Spleen/body mass (**A**) and thymus/body mass (**B**) ratios in the survived juvenile (3 weeks old) B6-1473C (saline-treated, n = 8; LPS-treated, n = 8) and B6-1473G (saline-treated, n = 8; LPS-treated, n = 8) males five days after single saline or LPS (2 mg/kg) ip administration. ** *p* < 0.01, *** *p* < 0.001 vs. saline-treated males of the same genotype; ## *p* < 0.01 vs. saline-treated B6-1474C males.

**Figure 19 biomolecules-15-00461-f019:**
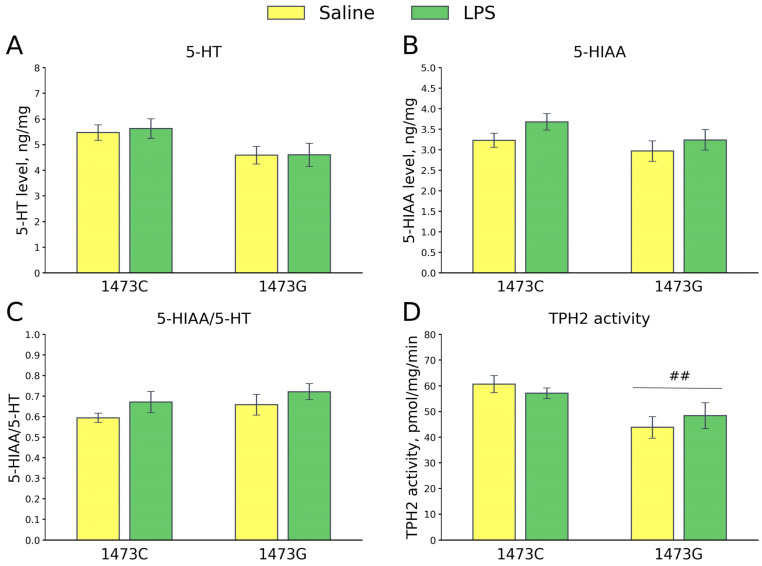
5-HT (**A**) and 5-HIAA (**B**) levels, 5-HIAA/5-HT ratios (**C**) and TPH2 activity (**D**) in the midbrains of survived juvenile B6-1473C (saline-treated, n = 8; LPS-treated, n = 8) and B6-1473G (saline-treated, n = 8; LPS-treated, n = 8) mice 5 days after saline or LPS (2 mg/kg) ip administration. ## *p* < 0.01 vs. B6-1473C mice.

**Figure 20 biomolecules-15-00461-f020:**
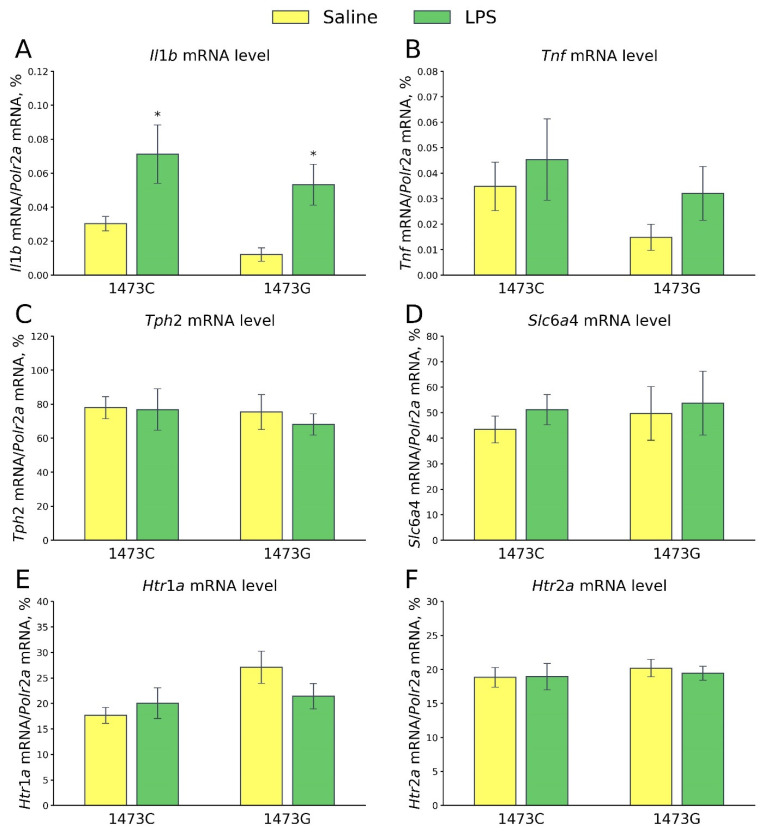
Levels of mRNA of *Il1*b (**A**), *Tnf* (**B**), *Tph2* (**C**), *Slc6a* (**D**), *Htr1a* (**E**) and *Htr2a* (**F**) genes in the midbrains of survived juvenile B6-1473C (saline-treated, n = 8; LPS-treated, n = 8) and B6-1473G (saline-treated, n = 8; LPS-treated, n = 8) mice 5 days after saline or LPS (2 mg/kg) ip administration. * *p* < 0.05 vs. saline-treated mice of the same genotype.

**Table 1 biomolecules-15-00461-t001:** Sequences, annealing temperatures of the primers and sizes of PCR products (amplicons).

Gene	Sequence	Annealing Temperature, °C	Amplicon Size, bp
Positive control	5′-TTTGACCCAAAGACGACCTGCTTGCA-3′ 5′-TGCATGCTTACTAGCCAACCATGAGACA-3′	62	523
*1473C specific*	5′-CAGAATTTCAATGCTCTGCGTGTGGG-3′	62	307
*1473G specific*	5′-CAGAATTTCAATGCTXTGCGTGTGGC-3′	62	307
*Polr2a*	5′-TGTGACAACTCCATACAATGC-3′ 5′-CTCTCTTACTGAATTTGCGTACT-3’	60	194
*Htr1a*	5′-GACTGCCACCCTCTGCCCTATATC-3′ 5′-TCAGCAAGGCAAACAATTCCAG-3′	62	200
*Htr2a*	5′-AGAAGCCACCTTGTGTGTGA-3′ 5′-TTGCTCATTGCTGATGGACT-3	61	169
*Tph2*	5′-CATTCCTCGCACAATTCCAGTCG-3′ 5′-AGTCTACATCCATCCCAACTGCTG-3′	62	239
*Tph1*	5′-ACAAACTCTACCCGACCCAC-3′ 5′- CAGTTGCGGGATGTTGTCTT-3′	63	197
*Slc6a4*	5′-CGCTCTACTACCTCATCTCCTCC-3′ 5′-GTCCTGGGCGAAGTAGTTGG-3′	63	101
*Il1b*	5′-GATCCCAAGCAATACCCAAA-3′ 5′-TAGAAACAGTCCAGCCCATAC-3′	61	226
*Tnf*	5′-AGCCGATGGGTTGTACCTTG-3′ 5′-GGTTGACTTTCTCCTGGTATGAGA-3′	59	211

**Table 2 biomolecules-15-00461-t002:** K_M_ and V_max_ values for BH_4_ and L-tryptophan for 1473C and 1473G forms of TPH2.

Trait	1473C	1473G	*p*
L-tryptophan			
V_max_	113.9 ± 2.6 pmol/mg/min	63.0 ± 2.2 pmol/mg/min	t_10_ = 21.1, *p* < 0.001
K_M_	20.3 ± 1.4 μM	11.0 ± 1.6 μM	t_10_ = 6.4, *p* < 0.001
BH_4_			
V_max_	74.6 ± 3.0 pmol/mg/min	54.7 ± 2.1 pmol/mg/min	t_12_ = 3.86, *p* = 0.002
K_M_	45.1 ± 4.6 μM	43.5 ± 4.3 μM	t_12_ = 0.35, *p* = 0.73

**Table 3 biomolecules-15-00461-t003:** Two-way ANOVA on the effect of “Genotype” and “Age” factors and their interaction on the variability of number and accumulated time (duration) of hind paws clasping in young and adult B6-1473C and B6-1473G males.

Trait	“Genotype”	“Age”	Interaction
Clasping frequency	F_1,85_ = 7.2, *p* = 0.009	F_1,85_ = 6.9, *p* = 0.01	F_1,85_ = 2.0, *p* = 0.16
Clasping duration	F_1,85_ = 6.5, *p* = 0.013	F_1,85_ = 5.9, *p* = 0.017	F_1,85_ = 2.2, *p* = 0.14

**Table 4 biomolecules-15-00461-t004:** Two-way ANOVA with repeated measures of variability in day distance travelled and sleep duration dynamics in survived juvenile B6-1473C and B6-1473G mice over five days after LPS administration.

	Distance Travelled	Sleep Duration
“Genotype”	F_1,25_ = 6.03, *p* = 0.022	F_1,25_ = 13.54, *p* = 0.001
“Treatment”	F_1,25_ = 2.02, *p* = 0.17	F_1,25_ = 2.47, *p* = 0.13
“Genotype” × “Treatment”	F_1,25_ = 0.49, *p* = 0.49	F_1,25_ = 1.52, *p* = 0.23
“Day”	F_4,100_ = 20.4, *p* < 0.001	F_4,100_ = 18.9, *p* < 0.001
“Day” × “Genotype”	F_4,100_ = 2.2, *p* = 0.07	F_4,100_ = 0.43, *p* = 0.78
“Day” × “Treatment”	F_4,100_ = 1.28, *p* = 0.28	F_4,100_ = 6.6, *p* < 0.001
“Day” × “Genotype” × “Treatment”	F_4,100_ = 1.7, *p* = 0.16	F_4,100_ = 2.46, *p* = 0.05

## Data Availability

The data presented in this study are available on request from the corresponding author. The data are not publicly available due to privacy or ethical restrictions.

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
