# Peer review of "The C1473G Mutation in the Mouse Tph2 Gene: From Molecular Mechanism to Biological Consequences†"

_biomolecules, 2025, doi:10.3390/biom15040461_

Round 1

Reviewer 1 Report

Comments and Suggestions for Authors

Biomolecules manuscript 3421224 "The C1473G mutation in the mouse Tph2 gene: from molecular mechanism to biological consequences" by N. V. Khotskin et al.

In this manuscript, the authors are interested in Tryptophan hydroxylase 2 (TPH2) mutations associated with psychopathology and resistance to antidepressant therapy and more specifically by the C1473G polymorphism that decreases TPH2 activity in the mouse brain. The authors used B6-1473C and B6-1473G congenic mice that differ only in the C to G substitution. They report that the C to G substitution reduces free energy of denaturation, stability and life time of TPH2 enzyme. They found that the 1473G allele causes hind legs dystonia in juvenile (3 weeks old) mice and a significant increase in the vulnerability to 2 mg/kg of bacterial lipopolysaccharide (LPS) in juvenile males. They found that a single ip LPS injection killed about 40% of young mutant, but not wild type mice.

This manuscript is sound and interesting for the reader, since the consequences of the C1473G polymorphism remained unclear. However, there are few adjustments that seems necessary before publication.

1-The authors report that the C to G substitution reduces stability and life time of TPH2 enzyme with a marked decrease in the TPH2 protein level in the midbrain of mice, but with no change in TPH2 activity in the whole brain of embryos.  The authors should document this apparent discrepancy further. In addition, they indicate that "a relatively high expression of the Tph1 gene in the brain of mouse embryos was revealed". Again, the readers need to know in more details this surprising observation, where this increase is taking place, and in which cells.

2-The authors report that the 1473G allele produces negative effect on survival of young mice with hind legs dystonia. The authors need again to document this observation in more details

3-The authors discuss that "Low TPH2 activity of the mutant enzyme does not seem to be sufficient restore the LPS-induced loss of 5-HT in the first 72 hours after LPS administration". We would need to have a more detailed experiment illustrating this point.

4-There are many typos. For example, line 566 what means "At the same time, no alteration in the brain 5-HT metabolism and the expression of Il1b, Tnf, Tph2, Slc6a4, Htr1a, Htr2a genes did not observed in the midbrain of the survived B6-1473G and B6-1473C mice”?

Comments on the Quality of English Language

Biomolecules manuscript 3421224 "The C1473G mutation in the mouse Tph2 gene: from molecular mechanism to biological consequences" by N. V. Khotskin et al.

In this manuscript, the authors are interested in Tryptophan hydroxylase 2 (TPH2) C1473G polymorphism that decreases TPH2 activity in the mouse brain. They found that the 1473G allele causes hind legs dystonia in juvenile (3 weeks old) mice and a significant increase in the vulnerability to 2 mg/kg of bacterial lipopolysaccharide (LPS) in juvenile males. This manuscript is sound and of interest, since the consequences of the C1473G polymorphism remained unclear. However, there are few additional information that seems necessary before publication.

Author Response

Biomolecules manuscript 3421224 "The C1473G mutation in the mouse Tph2 gene: from molecular mechanism to biological consequences" by N. V. Khotskin et al.

In this manuscript, the authors are interested in Tryptophan hydroxylase 2 (TPH2) mutations associated with psychopathology and resistance to antidepressant therapy and more specifically by the C1473G polymorphism that decreases TPH2 activity in the mouse brain. The authors used B6-1473C and B6-1473G congenic mice that differ only in the C to G substitution. They report that the C to G substitution reduces free energy of denaturation, stability and life time of TPH2 enzyme. They found that the 1473G allele causes hind legs dystonia in juvenile (3 weeks old) mice and a significant increase in the vulnerability to 2 mg/kg of bacterial lipopolysaccharide (LPS) in juvenile males. They found that a single ip LPS injection killed about 40% of young mutant, but not wild type mice.

This manuscript is sound and interesting for the reader, since the consequences of the C1473G polymorphism remained unclear. However, there are few adjustments that seems necessary before publication.

1-The authors report that the C to G substitution reduces stability and life time of TPH2 enzyme with a marked decrease in the TPH2 protein level in the midbrain of mice, but with no change in TPH2 activity in the whole brain of embryos.  The authors should document this apparent discrepancy further. In addition, they indicate that "a relatively high expression of the Tph1 gene in the brain of mouse embryos was revealed". Again, the readers need to know in more details this surprising observation, where this increase is taking place, and in which cells.

Answer. Unfortunately, at the moment we cannot explain the observed characteristics in the brain 5-HT system in 19 days old embryos (chapter 3.7). However, these results are not crucial for the paper and if the reviewer will insist, we can remove them from the results and discussion (the marked in yellow fragments together with Fig.12, Fig. 13 and corresponding references). The most informative result of this chapter is that the G allele produces a relatively small effect on the survival and body mass of 19 days old embryos.

2-The authors report that the 1473G allele produces negative effect on survival of young mice with hind legs dystonia. The authors need again to document this observation in more details

Answer. Of course it is only a hypothesis and we made corresponding corrected to his item.

3-The authors discuss that "Low TPH2 activity of the mutant enzyme does not seem to be sufficient restore the LPS-induced loss of 5-HT in the first 72 hours after LPS administration". We would need to have a more detailed experiment illustrating this point.

Answer. We agree that this phrase is rather unclear and we rewrote this item to make it more clear.

4-There are many typos. For example, line 566 what means "At the same time, no alteration in the brain 5-HT metabolism and the expression of Il1b, Tnf, Tph2, Slc6a4, Htr1a, Htr2a genes did not observed in the midbrain of the survived B6-1473G and B6-1473C mice”?

Answer. Thank you. We hired a professional proofreader to identify and remove all typos.

We thank very the reviewer for his/her valuable comments.

Reviewer 2 Report

Comments and Suggestions for Authors

The manuscript “ The C1473G mutation in the mouse Tph2 gene: from molecular mechanism to biological consequences" by Khotskin et al. is an experimental study that touches on the behavioral and biochemical consequences of the mutation in the gene encoding the THP2 enzyme. In general, the study is well-designed, and the presentation of the obtained results and the structure of the individual chapters are clear and comprehensive. However, I believe some points require clarification and should be discussed by the authors before this work can be considered for publication.

My first comments concern the chapter Materials and Methods. My first comments concern the chapter Materials and Methods. In my opinion, the authors should expand the descriptions of behavioral tests (sections: 2.2.4 and 2.2.6) as well as biochemical methods (sections: 2.4; 2.5.2 -Ë— 2.5.7) so that an interested person could repeat a given experimental procedure without referring to the cited works.

In the chapter Results, Section 3.2., Fig. 6C shows only representative blots from mutants 1373C and 1373G. Therefore, original gels with blots for all mice taken for calculations should be shown in Supplementary Materials.

In the descriptions of ANOVA results, please enter a uniform formula for presenting the results of this analysis, e.g. F (2,36) = 1.87, p = 0.17 instead of (F2,36) < 1 without specifying the p-value.

Please correct the descriptions of the results for which two-way ANOVA was used for analysis. Statistically significant differences between the tested groups are marked with symbols above particular bars only when an interaction between the tested parameters is found. When there is no interaction, only the effects of the individual parameters are given.

Author Response

The manuscript “ The C1473G mutation in the mouse Tph2 gene: from molecular mechanism to biological consequences" by Khotskin et al. is an experimental study that touches on the behavioral and biochemical consequences of the mutation in the gene encoding the THP2 enzyme. In general, the study is well-designed, and the presentation of the obtained results and the structure of the individual chapters are clear and comprehensive. However, I believe some points require clarification and should be discussed by the authors before this work can be considered for publication.

My first comments concern the chapter Materials and Methods. My first comments concern the chapter Materials and Methods. In my opinion, the authors should expand the descriptions of behavioral tests (sections: 2.2.4 and 2.2.6) as well as biochemical methods (sections: 2.4; 2.5.2 -Ë— 2.5.7) so that an interested person could repeat a given experimental procedure without referring to the cited works.

Answer. The first variant of this manuscript included the expanded descriptions of all methods used. However, the special program found a high percentage of auto-plagiarisms in its first version. The main source of this plagiarism was precisely the detailed descriptions of methods that are found in many of our other articles. In order to decrease the percentage of auto-plagiarisms, we restricted the descriptions of behavioral test and biochemical methods to references to our published papers. However, if the editors allow it and do not pay attention to auto-plagiarism, we can include a full description of all the methods used.

In the chapter Results, Section 3.2., Fig. 6C shows only representative blots from mutants 1373C and 1373G. Therefore, original gels with blots for all mice taken for calculations should be shown in Supplementary Materials.

Answer. OK, We put the original blots in the Supplement.

In the descriptions of ANOVA results, please enter a uniform formula for presenting the results of this analysis, e.g. F (2,36) = 1.87, p = 0.17 instead of (F2,36) < 1 without specifying the p-value.

Answer. OK. Corrected.

Please correct the descriptions of the results for which two-way ANOVA was used for analysis. Statistically significant differences between the tested groups are marked with symbols above particular bars only when an interaction between the tested parameters is found. When there is no interaction, only the effects of the individual parameters are given.

Answer. We agree with the reviewer and changed the Fig.17B and Fig.19D according to this comment. However, using a common line to label the effects of “Genotype” (Fig.14A,B) and “Treatment” (Fig.18A,B and Fig.20A) factors would make these figures difficult to understand. Therefore, these effects were labeled in these figures using asterisks above the corresponding bars.

We thank very much the reviewer for his/her valuable comments